# Iterative Circuit Repair Against Formal Specifications

**Matthias Cosler**
CISPA Helmholtz Center for Information Security
matthias.cosler@cispa.de

**Frederik Schmitt**
CISPA Helmholtz Center for Information Security
frederik.schmitt@cispa.de

**Christopher Hahn**
Stanford University
hahn@cs.stanford.edu

**Bernd Finkbeiner**
CISPA Helmholtz Center for Information Security
finkbeiner@cispa.de

## Abstract

We present a deep learning approach for repairing sequential circuits against formal specifications given in linear-time temporal logic (LTL). Given a defective circuit and its formal specification, we train Transformer models to output circuits that satisfy the corresponding specification. We propose a separated hierarchical Transformer for multimodal representation learning of the formal specification and the circuit. We introduce a data generation algorithm that enables generalization to more complex specifications and out-of-distribution datasets. In addition, our proposed repair mechanism significantly improves the automated synthesis of circuits from LTL specifications with Transformers. It improves the state-of-the-art by $6.8$ percentage points on held-out instances and $11.8$ percentage points on an out-of-distribution dataset from the annual reactive synthesis competition.

## 1 Introduction

Sequential circuit repair (Katz & Manna, 1975) refers to the task of given a formal specification and a defective circuit implementation automatically computing an implementation that satisfies the formal specification. Circuit repair finds application especially in formal verification. Examples are automated circuit debugging after model checking (Clarke, 1997) or correcting faulty circuit implementations predicted by heuristics such as neural networks (Schmitt et al., 2021b). In this paper, we design and study a deep learning approach to circuit repair for linear-time temporal logic (LTL) specifications (Pnueli, 1977) that also improves the state-of-the-art of synthesizing sequential circuits with neural networks.

We consider sequential circuit implementations that continuously interact with their environments. For example, an arbiter that manages access to a shared resource interacts with processes by giving out mutually exclusive grants to the shared resource. Linear-time temporal logic (LTL) and its dialects (e.g., STL Maler & Nickovic (2004) or CTL Clarke & Emerson (1981)) are widely used in academia and industry to specify the behavior of sequential circuits (e.g., Godhal et al. (2013); IEEE (2005); Horak et al. (2021)). A typical example is the response property $\Box(r \rightarrow \Diamond g)$, stating that it always ($\Box$) holds that request $r$ is eventually ($\Diamond$) answered by grant $g$. We can specify an arbiter that manages the access to a shared resource for four processes by combining response patterns for requests $r_0, \ldots, r_3$ and grants $g_0, \ldots, g_3$ with a mutual exclusion property as follows:

$$\Box(r_0 \rightarrow \Diamond g_0) \wedge \Box(r_1 \rightarrow \Diamond g_1) \wedge \Box(r_2 \rightarrow \Diamond g_2) \wedge \Box(r_3 \rightarrow \Diamond g_3) \qquad \textit{response properties}$$
$$\Box((\neg g_0 \wedge \neg g_1 \wedge (\neg g_2 \vee \neg g_3)) \vee ((\neg g_0 \vee \neg g_1) \wedge \neg g_2 \wedge \neg g_3)) \qquad \textit{mutual exclusion property}$$

A possible implementation of this specification is a circuit that gives grants based on a round-robin scheduler. However, running neural reactive synthesis (Schmitt et al., 2021b) on this specification results in a defective circuit as shown in Figure 1a. After model checking the implementation, we observe that the circuit is not keeping track of counting (missing an AND gate) and that the mutual exclusion property is violated (the same variable controls grants $g_0$ and $g_1$).

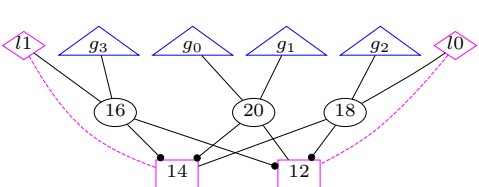

(a) Faulty circuit, predicted by synthesis model.

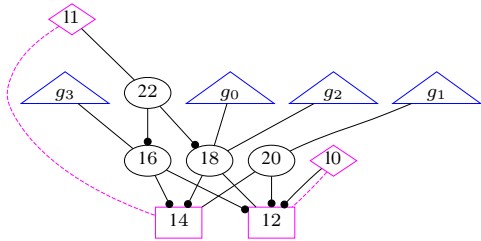

(b) Faulty circuit, first iteration of repair model.

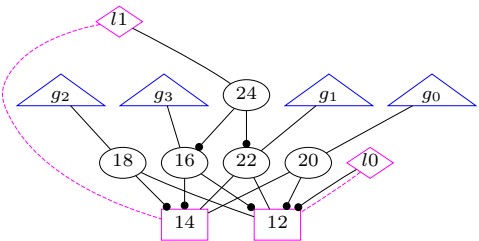

(c) Correct circuit. Final prediction of the repair model in the second iteration (DOT visualization on the left, model's output in AIGER on the right).

```
aag 12 5 2 5 5
2              input  0 (i0)
4              input  1 (r_2)
6              input  2 (r_0)
8              input  3 (r_3)
10             input  4 (r_1)
12 13          latch  0 (l0)
14 24          latch  1 (l1)
16             output 0 (g_3)
18             output 1 (g_2)
20             output 2 (g_0)
22             output 3 (g_1)
0              output 4 (o4)
16 15 13       and-gates
18 15 12       and-gates
20 14 13       and-gates
22 14 12       and-gates
24 23 17       and-gates
```

Figure 1: Circuit representations of 4-process arbiter implementation in DOT visualizations: The triangles represent inputs and outputs, the rectangles represent variables, the diamond-shaped nodes represent latches (flip-flop), ovals represent AND gates, and the black dots represent inverter (NOT gates). The output of our repair model is given as an AIGER circuit (bottom right).

We present the first deep learning approach to repair such faulty circuits, inspired by the successful application of deep learning to the LTL trace generation (Hahn et al., 2021) and reactive synthesis problem (Schmitt et al., 2021b). We introduce a new Transformer architecture, the *separated hierarchical Transformer*, that accounts for the different characteristics of the problem's input. The separated hierarchical Transformer combines the advantages of the hierarchical Transformer (Li et al., 2021) with the multimodal representation learning of an LTL specification and a faulty circuit. In particular, it utilizes that LTL specifications typically consist of reoccurring patterns. This architecture can successfully be trained on the circuit repair problem. Our model, for example, produces a correct circuit implementation of the round-robin strategy by repairing the faulty circuit in Figure 1a in only two iterations. Each iteration predicts a circuit based on the specification and a faulty circuit as input. The result of the first iteration is shown in Figure 1b. The circuit remains faulty, with two of the four grants still controlled by the same variable. Progress was made, however, towards a functioning counter: latch $l1$ now consists of a combination of AND gates and inverters expressive enough to represent a counter. The second iteration finally results in a correct implementation, as shown in Figure 1c.

To effectively train and enable further research on repair models, we provide open-source datasets and our open-source implementation for the supervised training of the circuit repair problem[1]. We demonstrate that the trained separated hierarchical Transformer architecture generalizes to unseen specifications and faulty circuits. Further, we show that our approach can be combined with the existing neural method for synthesizing sequential circuits (Schmitt et al., 2021b) by repairing its mispredictions, improving the overall accuracy substantially. We made a significant improvement of 6.8 percentage points to a total of $84\%$ on held-out-instances, while an even more significant improvement was made on out-of-distribution datasets with 11.8 percentage points on samples from the annual reactive synthesis competition SYNTCOMP (Jacobs et al., 2022a).

---

[1] https://github.com/reactive-systems/circuit-repair

## 2 RELATED WORK

**Circuit repair.** The repair problem is an active field of research dating back to Katz & Manna (1975). Jobstmann et al. (2005; 2012) show a game-based approach to repair programs using LTL specifications. Baumeister et al. (2020) propose an approach for synthesizing reactive systems from LTL specifications iteratively through repair steps. Ahmad et al. (2022) presented a framework for automatically repairing defects in hardware design languages like Verilog. Staber et al. (2005) combine fault localization and correction for sequential systems with LTL specifications.

**Deep learning for temporal logics and hardware.** Hahn et al. (2021); Schmitt et al. (2021a) initiated the study of deep learning for temporal logics; showing that a Transformer can understand the semantics of temporal and propositional logics. Schmitt et al. (2021b) successfully applied the Transformer to the reactive synthesis problem. Kreber & Hahn (2021) showed that (W)GANs equipped with Transformer encoders can generate sensible and challenging training data for LTL problems. Luo et al. apply deep learning to the $LTL_f$ satisfiability problem. Mukherjee et al. (2022) present a deep learning approach to learn graph representations for LTL model checking. Vasudevan et al. (2021) applied deep learning to learn semantic abstractions of hardware designs. In Hahn et al. (2022), the authors generate formal specifications from unstructured natural language.

**Reactive synthesis.** The hardware synthesis problem traces back to Alonzo Church in 1957 (Church, 1963). Buchi & Landweber (1990) provided solutions, although only theoretically, already in 1969. Since then, significant advances in the field have been made algorithmically, e.g., with a quasi-polynomial algorithm for parity games (Calude et al., 2020), conceptually with distributed (Pneuli & Rosner, 1990) and bounded synthesis (Finkbeiner & Schewe, 2013), and on efficient fragments, e.g., GR(1) (Piterman et al., 2006) synthesis. Synthesis algorithms have been developed for hyperproperties (Finkbeiner et al., 2020). Recently, deep learning has been successfully applied to the hardware synthesis problem (Schmitt et al., 2021b). Compared to classical synthesis, a deep learning approach can be more efficient, with the tradeoff of being inherently incomplete. The field can build on a rich supply of tools (e.g. (Bohy et al., 2012; Faymonville et al., 2017; Meyer et al., 2018a)). A yearly competition (SYNTCOMP) (Jacobs & Pérez) is held at CAV.

## 3 DATASETS

We build on the reactive synthesis dataset from Schmitt et al. (2021b), where each sample consists of *two* entries: a formal specification in LTL and a target circuit that implements the specification given in the AIGER format. We construct a dataset for the circuit repair problem that consists of *three* entries: a formal specification in LTL, a defective circuit, and the corrected target circuit. In Section 3.1, we give details of the domain-specific languages for the circuit repair problem's input. Section 3.2 describes the data generation process and summarizes the dataset that resulted in the best-performing model (see Section 6 for ablations). We approach the generation of the dataset from two angles: 1) we collect mispredictions, i.e., faulty circuits predicted by a neural model, and 2) we introduce semantic errors to correct circuits in a way that they mimic human mistakes.

### 3.1 LINEAR-TIME TEMPORAL LOGIC (LTL) AND AND-INVERTER GRAPHS (AIGER)

**LTL specifications.** The specification consists of two lists of sub-specifications: *assumptions* and *guarantees*. Assumptions pose restrictions on the environment behavior, while guarantees describe how the circuit has to react to the environment. They jointly build an LTL specification as follows: $spec := (assumption_1 \wedge \cdots \wedge assumption_n) \to (guarantee_1 \wedge \cdots \wedge guarantee_m)$. A specification is called *realizabile* if there exists a circuit implementing the required behavior and called *unrealizable* if no such circuit exists. For example, an implementation can be unrealizable if there are contradictions in the required behavior, or if the environment assumptions are not restrictive enough. Formally, an LTL specification is defined over traces through the circuit. A circuit $C$ *satisfies* an LTL specification $\varphi$ if all possible traces through the circuit $Traces_C$ satisfy the specification, i.e., if $\forall t \in Traces_C.\, t \models \varphi$. For example, the LTL formula $\Box(\neg g_0 \vee \neg g_1)$ from the arbiter example in Section 3.1 requires all traces through the circuit to respect mutual exclusive behavior between $g_0$ and $g_1$. If a specification is realizable, the target circuit represents the implementation, and if a specification is unrealizable, the target circuit represents the counter-strategy of the environment showing that no such implementation exists. The formal semantics of LTL can be found in Appendix A.

---

**Algorithm 1** Algorithm for introducing errors to correct circuit implementations.

---

1: **Input**: circuit $C$, number of changes standard deviation $\sigma_c$, maximum number of changes $M_c$, new variable number standard deviation $\sigma_v$, delete line probability $p_{delete}$
2: **Output**: circuit $C$

3: $changes \sim \mathcal{N}^d(0, \sigma_c{}^2, 1, M_c)$         ▷ Sample from discrete truncated Gaussian
4: **for** $i = 0$ to $changes$ **do**
5:     **with** probability $p_{delete}$ **do**
6:        $l \sim \mathcal{U}(1, \text{number of lines in } C)$        ▷ Sample line number uniformly
7:        Remove line $l$ of $C$
8:     **else**
9:        $pos \sim \mathcal{U}(1, \text{number of positions in } C)$        ▷ Sample position uniformly
10:        $var' \leftarrow var \leftarrow$ variable number at position $pos$ in $C$
11:        **while** $var = var'$ **do**
12:           $var' \sim \mathcal{N}^d(var, \sigma_v{}^2, 0, 61)$        ▷ Sample from discrete truncated Gaussian
13:        replace variable number at position $pos$ in $C$ with $var'$

---

**AIGER format.** The defective and target circuits, are in a text-based representation of And-Inverter Graphs called AIGER Format (Brummayer et al., 2007); see, for example, bottom-right of Figure 1. A line in the AIGER format defines nodes such as latches (flip-flops) and AND-gates by defining the inputs and outputs of the respective node. Connections between nodes are described by the variable numbers used as the input and output of nodes. A latch is defined by one input and one output connection, whereas two inputs and one output connection define an AND gate. Inputs and outputs of the whole circuit are defined through lines with a single variable number that describes the connection to a node. The parity of the variable number implicitly gives negations. Hence, two consecutive numbers describe the same connection, with odd numbers representing the negated value of the preceding even variable number. The numbers 0 and 1 are the constants *False* and *True*. The full definition of AIGER circuits can be found in Appendix B.

## 3.2 DATA GENERATION

We replicated the neural circuit synthesis model of Schmitt et al. (2021b) and evaluated the model with all specifications from their dataset while keeping the training, validation, and test split separate. We evaluated with a beam size of 3, resulting in a dataset `RepairRaw` of roughly $580\,000$ specifications and corresponding (possibly faulty) circuits in the training split and about $72\,000$ in the validation and test split, respectively. We model-check each predicted circuit against its specification with nuXmv (Cavada et al., 2014) to classify defective implementations into the following classes. A sample is *violated* if the predicted circuit is defective, i.e., violates the specification (55%). A sample is *matching* if the prediction of the synthesis model is completely identical to the target circuit in the dataset (16%). Lastly, a sample is *satisfied* when the predicted circuit satisfies the specification (or represents a correct counter-strategy) but is no match (29%), which regularly happens as a specification has multiple correct implementations. For example, consider our round-robin scheduler from the introduction: the specification does not specify the order in which the processes are given access to the resource.

We construct our final dataset from `RepairRaw` in two steps. In the first step, we consider the violating samples, i.e., mispredictions of the neural circuit synthesis network, which are natural candidates for a circuit repair dataset. In the second step, we introduce mistakes inspired by human errors into correct implementations (see Figure 6 in the appendix for an overview of the dataset generation and its parameters). In the following, we describe these steps in detail.

**Mispredictions of neural circuit synthesis.** We first consider the violating samples from `RepairRaw`. Likewise to a specification having multiple correct implementations, a defective circuit has multiple possible fixes, leading to correct yet different implementations. For a given defective circuit, a fix can thus either be small and straightforward or lengthy and complicated. In a supervised learning setting, this leads us to the issue of *misleading target circuits*. This concerns samples where only a lengthy and complicated fix of the faulty circuit leads to the target circuit,

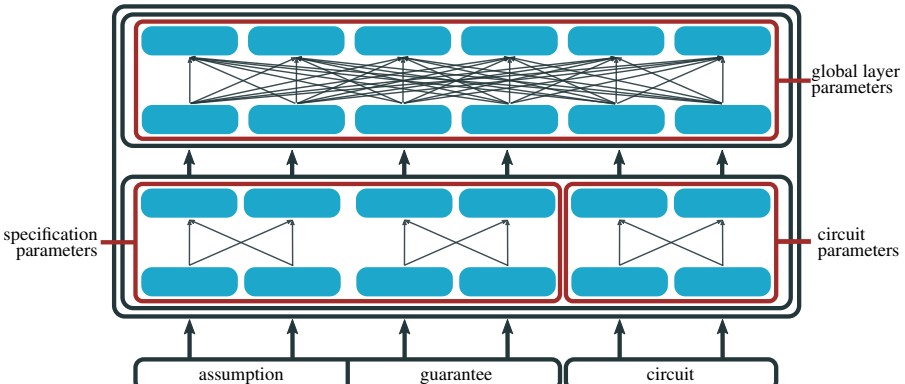

Figure 2: The structure of global and local layers in the separated hierarchical Transformer. For simplicity, shown for a single assumption, a single guarantee and only two tokens each.

although a minor fix would also lead to a correct but different implementation. We identify misleading targets by searching for alternative solutions with the synthesis model (up to a beam size of 4). If the model finds a correct alternative circuit with a smaller Levenshtein distance (see Appendix C for a definition) to the faulty circuit, a fix leading to the alternative circuit is smaller than a fix to the original target. The target circuit will be replaced accordingly with the alternative circuit. We select all samples with a Levenshtein distance to the target circuit $\leq 50$ for the final dataset.

**Introducing errors.** We propose Algorithm 1, which probabilistically introduces human-like mistakes into correct circuits. Such mistakes include missing latches, AND gates, or inverters and miswired connections between the components. First, we determine the number of mistakes or changes we will introduce to the circuit. For that, we sample from a discrete truncated normal distribution around zero, with a standard deviation of $7.5$ and bounds from $1$ to $50$. For each change, we flip a coin with the probability of $p_{delete} = 0.2$ for deleting a line from the AIGER circuit and $1 - p_{delete} = 0.8$ for changing a variable number. For *deleting*, we uniformly choose a line from the AIGER circuit to remove. We do not remove inputs or outputs to stay consistent with the dataset. For *changing* a variable number, we uniformly choose a position of a variable number. The position can be an input, output, inbound edge(s), or outbound edge of a latch or AND gate. We replace the variable number at this position with a new variable number that is determined by sampling a discrete truncated normal distribution around the old variable number, a standard deviation of $10$, and bounds given by the minimal and maximal possible variable number in the dataset ($0$ to $61$). The new variable number cannot be the mean itself to ensure a definite change. For a visualization of the discrete truncated normal distributions, see Figure 7 in the appendix. Lastly, we spot-check altered circuits by model-checking to determine whether introduced changes create a faulty circuit. Only in less than $2\%$ of the cases the circuit still satisfies the specification.

**Final Dataset.** In the final dataset `Repair`, $61\%$ of the samples contain circuits with errors introduced by Algorithm 1, while the others are based on mispredicted circuits. In $38\%$ of cases, the samples have a Levenshtein distance of less than $10$ between the repair circuit and the target circuit. In total, the Levenshtein distance in the dataset has a mean of $15.7$ with a standard deviation of $12.77$, and the median is at $13$ (see Figure 8 in Appendix D for its composition).

## 4 ARCHITECTURE

In this section, we introduce the separated hierarchical Transformer architecture, a variation of the hierarchical Transformer Li et al. (2021), and provide further details on our architectural setup. The hierarchical Transformer has been shown to be superior to a vanilla Transformer in many applications including logical and mathematical problems Li et al. (2021); Schmitt et al. (2021b). The hierarchical Transformer, as well as the novel separated hierarchical Transformer, is invariant against the order of the assumptions and guarantees in the specification.

### 4.1 SEPARATED HIERARCHICAL TRANSFORMER

The encoder of a hierarchical Transformer contains two types of hierarchically structured layers. *Local* layers only see parts of the input, while *global* layers handle the combined output of all local layers. Contrary to the original Transformer, the input is partitioned before being fed into the local layers. A positional encoding is applied separately to each partition of the input. Model parameters are shared between the local layers, but no attention can be calculated between tokens in different partitions. The hierarchical Transformer has been beneficial to understanding repetitive structures in mathematics (Li et al., 2021) and has shown to be superior for processing LTL specifications (Schmitt et al., 2021b).

We extend the hierarchical Transformer to a separated hierarchical Transformer, which has two types of local layers: Each *separated local layer* is an independent encoder; therefore, separated local layers do not share any model parameters. Attention calculations are done independently for each local layer. A visualization of the proposed Architecture is shown in Figure 2. *Shared local layers* are identical to local layers in the hierarchical Transformer. A separated local layer contains one or more shared local layers. The results of the separated and shared local layers are concatenated and fed into the global layer. While the number of shared local layers does not change the model size, multiple separated local layers introduce slightly more model parameters. The separated hierarchical Transformer handles multiple independent inputs that differ in structure, type, or length better.

### 4.2 ARCHITECTURAL SETUP

We separate the specification and the faulty circuit with a separate local layer. The specification is partitioned into its guarantees and assumptions, which we feed into shared local layers. Let *Attention* be the attention function of Vaswani et al. (2017) (see Appendix K). When identifying the assumptions $assumption_1, \cdots, assumption_n$ and guarantees $guarantee_1, \cdots, guarantee_m$ with specification properties $p_1, \ldots, p_{n+m}$, the following computations are performed in a shared local layer:

$$Attention(H_{p_i}W_S^Q, H_{p_i}W_S^K, H_{p_i}W_S^V) \quad \text{where} \quad p_i \in \{p_1, \ldots, p_{n+m}\} \ ,$$

where $H_{p_i}$ denotes the stacked representations of all positions of specification property $p_i$. Therefore, the attention computation is limited between tokens in each guarantee and between tokens in each assumption while the learned parameters $W_S^Q, W_S^K, W_S^V$ are shared between all guarantees and assumptions. The separated local layer that processes the circuit performs the attention computation:

$$Attention(H_C W_C^Q, H_C W_C^K, H_C W_C^V) \ ,$$

where $H_C$ denotes the stacked representations of all positions of the circuit. Therefore, the computation is performed over all tokens in the circuit but the parameters $W_C^Q, W_C^K, W_C^V$ are different from the parameters for the specification (see Figure 2).

For embedding and tokenization, we specialize in the Domain Specific Language (DSL) of LTL formulas and AIGER circuits with only a few symbols. For every symbol in the DSL, we introduce a token. Variables in properties (i.e., assumptions and guarantees) are limited to five inputs $i_0 \cdots i_4$ and five outputs $o_0 \cdots o_4$, for each of which we introduce a token. In the AIGER format (used for the faulty circuit and the target circuit), we fix the variable numbers to the range of 0 to 61, thereby indirectly limiting the size of the circuit, while allowing for reasonable expressiveness. We set a special token as a prefix to the circuit embedding to encode the presumed realizability of the specification. This determines whether the circuit represents a satisfying circuit or a counter strategy. We embed the tokens by applying a one-hot encoding which we multiply with a learned embedding matrix. Properties have a tree positional encoding (Shiv & Quirk, 2019) as used for LTL formulas by (Hahn et al., 2021). This encoding incorporates the tree structure of the LTL formula into the positional encoding and allows easy calculations between tree relations. For circuits, we use the standard linear positional encoding from Vaswani et al. (2017).

## 5 EXPERIMENTS

In this section, we report on experimental results. We first describe our training setup in Section 5.1 before evaluating the model with two different methods. The *model evaluation* shows the evaluation

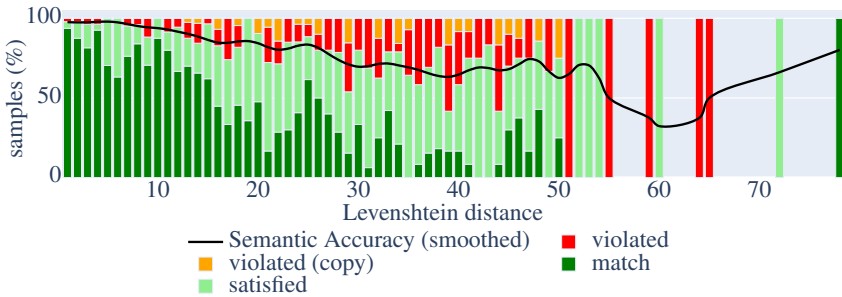

Figure 3: Accuracy broken down by the Levenshtein distance between faulty and target circuit.

of the repair model on the `Repair` dataset distribution(Section 5.2). In the *synthesis pipeline evaluation*, the repair model is evaluated on the predictions of the synthesis model and then repeatedly evaluated on its predictions (Section 5.3). We differentiate between syntactic and semantic accuracy, following Hahn et al. (2021). A sample is considered semantically correct if the prediction satisfies the specification. We consider the prediction syntactically correct if it is identical to the target.

## 5.1 TRAINING SETUP

We trained a separated hierarchical Transformer with $4$ heads in all attention layers, $4$ stacked local layers in both separated local layers, and $4$ stacked layers in the global layer. The decoder stack contains $8$ stacked decoders. The embedding size in the decoder and encoder is $256$ and all feed-forward networks have a size of $1024$ and use the Rectified Linear Unit (ReLU) activation function. We use the Adam optimizer (Kingma & Ba, 2017) with $\beta_1 = 0.9, \beta_2 = 0.98$ and $\epsilon = 10^{-9}$ and $4000$ warmup steps with an increased learning rate, as proposed in Vaswani et al. (2017). We trained on an single GPU of a NVIDIA DGX A100 system with a batch size of $256$ for $20\,000$ steps. We restricted the specification input to $5$ inputs and $5$ outputs, no more than $12$ properties (assumptions + guarantees) and no properties of a size of its abstract syntax tree (AST) greater than $25$.

## 5.2 MODEL EVALUATION

We evaluated the model up to a beam size of $16$. The key results of the model evaluation can be found at the top of Table 1. With a beam size of $16$, the model outputs a correct implementation in $84\%$ of the cases on a single try. When analyzing the beams, we found that the model shows enormous variety when fixing the circuits. Almost half of the beams result in correct implementations. To investigate if the model performs a repair operation, we identify samples where the model copied the defective circuit (*Violated (Copy)*). The model only copied $31$ of $1024$ samples. We, additionally, track if the predicted circuit contains syntax errors, which rarely happens (a total of $8$ errors out of every beam). We provide insight into the model's performance by analyzing a) what exactly makes a sample challenging to solve for the model and b) if the model makes significant improvements towards the target circuit even when the prediction violates the specification.

**Difficulty measures.** We consider three parameters to measure the difficulty of solving a specific repair problem: the size of the specification (the LTL formula's AST), the size of the target circuit (AND gates + latches), and the Levenshtein distance between the defective circuit and the target circuit. The Levenshtein distance is the dominant indicator of a sample's difficulty (see Figure 3). However, the specification and circuit size is, perhaps surprisingly, less of a factor (see Figure 11 and Figure 10 in the appendix). This indicates that our approach has the potential to scale up to larger circuits when increasing the model size.

**Improvement measures.** We semantically and syntactically approximate whether a violating prediction is still an improvement over the faulty input circuit. For syntactic improvement, we calculate the difference between the distance of the faulty input and target circuit $lev(C_i, C_t)$ and the distance between prediction and target circuit $lev(C_p, C_t)$. If the difference is below zero: $lev(C_p, C_t) - lev(C_f, C_t) < 0$, the model syntactically improved the faulty circuit towards the target circuit. On our test set, violated circuits improved by $-9.98$ edits on average. For semantic improvement, we obtained a set of sub-specifications by creating a new specification with each

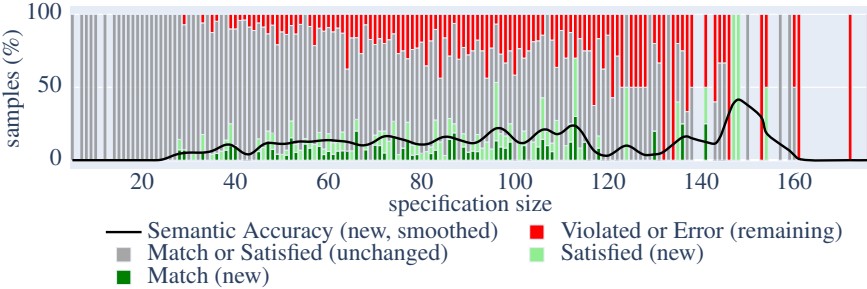

Figure 4: Improvements on the reactive synthesis held-out test set (see `test` in Table 1) broken down by the size of the specifications AST. We aggregate the best result from all iterations over 16 beams. The annotations *new* and *unchanged* indicate whether the status improved from the evaluation of the synthesis model to the evaluation of the repair model.

guarantee from the original specification: Let $a_1$ to $a_n$ be the original assumptions and $g_1$ to $g_m$ the original guarantees, the set of sub-specifications is defined as $\{(a_1 \wedge \cdots \wedge a_n) \rightarrow g_i \mid 1 \le i \le m\}$. We approximate that, the more sub-specifications a circuit satisfies, the closer it is semantically to a correct circuit. On our test set, in $75.9\%$, the prediction satisfied more sub-specifications, and in $2.4\%$, the prediction satisfied fewer sub-specifications. For a more detailed insight, we supported violin plots for syntactic (Figure 12) and semantic (Figure 13) improvement in the Appendix. Since in both approximations, even violating predictions are an improvement over the faulty input, this poses the natural question if the model's performance can be increased by iteratively querying the model on its predictions. In the next section, we investigate this more in-depth by applying our repair model iteratively to the prediction of a neural circuit synthesis model including real-world examples.

## 5.3 SYNTHESIS PIPELINE EVALUATION

We demonstrate how our approach can be used to improve the current state-of-the-art for neural reactive synthesis (see Figure 5). We first evaluate the synthesis model we replicated from Schmitt et al. (2021b). If the predicted circuit violates the specification, we feed the specification together with the violating circuit into our repair model. If the prediction still violates the specification after applying the repair model once, we re-feed the specification with the new violating circuit into the repair model until it is solved. Using the presented pipeline, we improve the results of Schmitt et al. (2021b) significantly, as shown in the bottom half of Table 1. We evaluate held-out samples from the synthesis dataset and out-of-distribution benchmarks and filtered out samples that exceed our input restrictions (see Section 5.1). The datasets `test` contain randomly sampled held-out instances from the repair and neural synthesis dataset, respectively. Figure 4 shows an in-depth analysis of the status changes of the samples when applying the synthesis pipeline.

Table 1: Syntactic and semantic accuracy of the model (top) and pipeline (bottom) evaluation.

| Beam Size | 1 | 16 |
|---|---|---|
| semantic accuracy | 58.3% | 84.8% |
| syntactic accuracy | 29.4% | 53.2% |
| correct beams per sample | 0.58 | 6.57 |

| | synthesis model | after first iteration | after up to $n$ iterations | $n$ |
|---|---|---|---|---|
| test (repair dataset) | - | 84.2% | 87.5% (+3.3) | 5 |
| test (synthesis dataset) | 77.1% | 82.6% (+5.5) | 83.9% (+6.8) | 5 |
| timeouts | 26.1% | 32.5% (+7.4) | 34.2% (+8.1) | 5 |
| syntcomp | 64.1% | 71.7% (+7.6) | 75.9% (+11.8) | 2 |
| smart home | 42.9% | 61.9% (+19) | 66.7% (+23.8) | 2 |

Light and dark green represent the instances that were additionally solved with our repair approach; gray represent the instances that were already initially solved by the synthesis network. The problem becomes increasingly more challenging with a larger target circuit size. In total, we achieve an improvement of 6.8 percentage points. To show that our improvement over the state-of-the-art is not due to scaling but rather a combination of new training data, architecture and iterative evaluation, we additionally scaled the model from Schmitt et al. (2021b) to match or exceed the number of parameters of our model. The parameter-matched models only lead to insignificant improvements over the base model (see Table 2 in the Appendix). We further identified a set of held-out samples were our approach performs significantly better than the classical state-of-the-art synthesizer tool Strix (Meyer et al., 2018b): Samples in `timeouts` could not been solved by Strix in 1h, of which we still achieve 34.2% with an improvement of 8.1 percentage points. Even more significant improvement can be observed in real-world samples from the annual synthesis competitions and out-of-distribution benchmarks: The dataset `smart home` are benchmarks for synthesizing properties over smart home applications (Geier et al., 2022), where we improve by 11.8 percentage points. The dataset `syntcomp` contains benchmarks from the annual reactive synthesis competition (Jacobs et al., 2022a;b), where the model pipeline improves the state-of-the-art by 23.8 percentage points and even by 19 percentage points by applying it once.

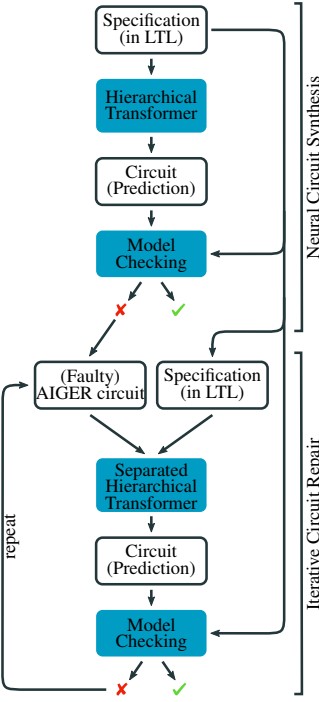

Figure 5: Pipeline structure

## 6  ABLATIONS

We performed various ablation studies that the interested reader can find in the appendix. In particular, we parameterized our data generation for constructing the circuit repair dataset (see Figure 6 in Appendix D). An extensive collection of over 100 generated datasets are available through our code at GitHub[2]. We trained various models based on these datasets and different hyperparameters, also available at GitHub. A hyperparameter study can be found in Figure 3 in Appendix I. An in-depth analysis of the results of different models tested in the synthesis pipeline can be found in Appendix J.

## 7  CONCLUSION

In this paper, we studied the first application of neural networks to the circuit repair problem. We introduced the separated hierarchical Transformer to account for the multimodal input of the problem. We provided a data generation method with a novel algorithm for introducing errors to circuit implementations. A separated hierarchical Transformer model was successfully trained to repair defective sequential circuits. The resulting model was used to significantly improve the state-of-the-art in neural circuit synthesis. Additionally, our experiments indicate that the separated hierarchical Transformer has the potential to scale up to even larger circuits.

Our approach can find applications in the broader hardware verification community. Possible applications include the automated debugging of defective hardware after model checking or testing. Due to its efficiency, a well-performing neural repair method reduces the necessary human interaction in the hardware design process. The benefit of a deep learning approach to the circuit repair problem is the scalability and generalization capabilities of deep neural networks: this allows for an efficient re-feeding of faulty circuits into the network when classical approaches suffer from the problem's high computational complexity. Moreover, neural networks generalize beyond classical repair operations, whereas classical approaches are limited in their transformations, such as the limitation of replacing boolean functions. Future work includes, for example, the extension of our approach to hardware description languages, such as VHDL or Verilog, and the extension to other specification languages that express security policies, such as noninterference or observational determinism.

---

[2]https://github.com/reactive-systems/circuit-repair

## REPRODUCIBILITY STATEMENT

All models, datasets, code, and guides are available in the corresponding code repository. All our datasets and models mentioned in the paper, the code of the data generation method, and the code for training new models as well as evaluating existing models are licensed under the open-source MIT License. Multiple Jupyter notebooks guide the interested reader through the use of the code to allow low-effort reproducibility of all our results and encourage fellow researchers to use, extend and build on our work.

## ACKNOWLEDGMENTS

This work was partially supported by the European Research Council (ERC) Grant HYPER (No. 101055412).

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

# A  LINEAR-TIME TEMPORAL LOGIC (LTL)

The syntax of linear-time temporal logic (LTL) Pnueli (1977) is given as follows.

$$\varphi := p \mid \varphi \wedge \varphi \mid \neg \varphi \mid \bigcirc \varphi \mid \varphi \, \mathcal{U} \, \varphi \ ,$$

where $p$ is an atomic proposition $p \in AP$. In this context, we assume that the set of atomic propositions $AP$ can be partitioned into inputs $I$ and outputs $O$: $AP = I \dot{\cup} O$.

The semantics of LTL is defined over a set of traces: $TR := (2^{AP})^{\omega}$. Let $\pi \in TR$ be trace, $\pi_{[0]}$ the starting element of a trace $\pi$ and for a $k \in \mathbb{N}$ and be $\pi_{[k]}$ be the kth element of the trace $\pi$. With $\pi_{[k,\infty]}$ we denote the infinite suffix of $\pi$ starting at $k$. We write $\pi \models \varphi$ for *the trace $\pi$ satisfies the formula $\varphi$*.

For a trace $\pi \in TR$, $p \in AP$ and formulas $\varphi$ the semantics of LTL is defined as follows:

- $\pi \models \neg \varphi$ iff $\pi \not\models \varphi$
- $\pi \models p$ iff $p \in \pi_{[0]}$ ; $\pi \models \neg p$ iff $p \notin \pi_{[0]}$
- $\pi \models \varphi_1 \wedge \varphi_2$ iff $\pi \models \varphi_1$ and $\pi \models \varphi_2$
- $\pi \models \bigcirc \varphi$ iff $\pi_{[1]} \models \varphi$
- $\pi \models \varphi_1 \, \mathcal{U} \, \varphi_2$ iff $\exists l \in \mathbb{N} : (\pi_{[l,\infty]} \models \varphi_2 \wedge \forall m \in [0, l-1] : \pi_{[m,\infty]} \models \varphi_1)$  .

We use further temporal and boolean operators that can be derived from the ones defined above. That includes $\vee, \rightarrow, \leftrightarrow$ as boolean operators and the following temporal operators:

- $\varphi_1 \, \mathcal{R} \, \varphi_2$ *(release)* is defined as $\neg(\neg\varphi_1 \, \mathcal{U} \, \neg\varphi_2)$
- $\Box \varphi$ *(globally)* is defined as *false* $\mathcal{R} \, \varphi$
- $\Diamond \varphi$ *(eventually)* is defined as *true* $\mathcal{U} \, \varphi$  .

### REACTIVE SYNTHESIS

Reactive synthesis is the task to find a circuit $C$ that satisfies a given formal specification $\varphi$, i.e., $\forall t \in \text{Traces}_C. \ t \models \varphi$, or determine that no such circuit exists. We consider formal specifications that are formulas over a set of atomic propositions ($AP$) in LTL. The specification defines the desired behavior of a system based on a set of input and output variables. As the system, we consider circuits, more precisely a text representation of And-Inverter Graphs, called AIGER circuits. And-Inverter Graphs connect input and output edges using AND gates, NOT gates (inverter), and memory cells (latches).

# B  AND-INVERTER GRAPHS

And-Inverter Graphs are graphs that describe hardware circuits. The graph connects input edges with output edges through AND gates, latches, and implicit NOT gates. We usually represent this graph by a text version called the AIGER Format Brummayer et al. (2007). The AIGER format uses variable numbers that define variables. Variables can be interpreted as wired connections in a circuit or as edges in a graph, where gates and latches are nodes.

- A negation is implicitly encoded by distinguishing between even and odd variable numbers. Two successive variable numbers represent the same variable, the even variable number represents the non-negated variable, and the odd variable number represents the negated variable. The variable numbers `0` and `1` have the constant values `FALSE` and `TRUE`.
- Each input and output edge is defined by a single variable number, respectively.
- An AND gate is defined by three variable numbers. The first variable number defines the outbound edge of the AND gate, and the following two variable numbers are inbound edges. The value of the outbound variable is determined by the conjunction of the values of both inbound variables.

- A latch is defined by two variable numbers: an outbound edge and an inbound edge. The value of the outbound variable is the value of the inbound variable at the previous time step. In the first time step, the outbound variable is initialized as `FALSE`.

The AIGER format starts with a header, beginning with the letters `aag` and following five non-negative integers `M, I, L, O, A` with the following meaning:

```
M=maximum variable index
I=number of inputs
L=number of latches
O=number of outputs
A=number of AND gates
```

After the header, each line represents a definition of either input, latch, output, or AND gate in this order. The numbers in the header define the number of lines associated with each type. After the definition of the circuit, an optional symbol table might follow, where we can define names for inputs, outputs, latches, and AND gates. In this context, the circuit can either describe a satisfying system or a counter strategy to the specification.

## C  LEVENSHTEIN DISTANCE

The Levenshtein distance is an edit distance metric, measuring the degree of distinction between two strings. Let $s_1$ and $s_2$ two given strings, then the Levenshtein distance $lev(s_1, s_2)$ is a the number of actions necessary to transform string $s_1$ into string $s_2$ or vice versa. Possible actions are deletions, insertions, and substitutions.

## D  DATA GENERATION

In Figure 6 we sketch the data generation process. The base of the process is the evaluation of a model for neural circuit synthesis. This is parameterized as multiple beam size are possible. For mispredicted samples, we replace misleading targets (see Section 3.2). This is optional but our experiments showed that the training benefits from this step. Up to a given Levenshtein distance, we collect samples for the final dataset. All other samples (greater Levenshtein distances and correct predictions) are processed in Section 3.2 and Algorithm 1. This process is optional, can be applied to only some samples and is also parameterized. The results can be used for the final dataset or are, depending on various parameters, discarded.

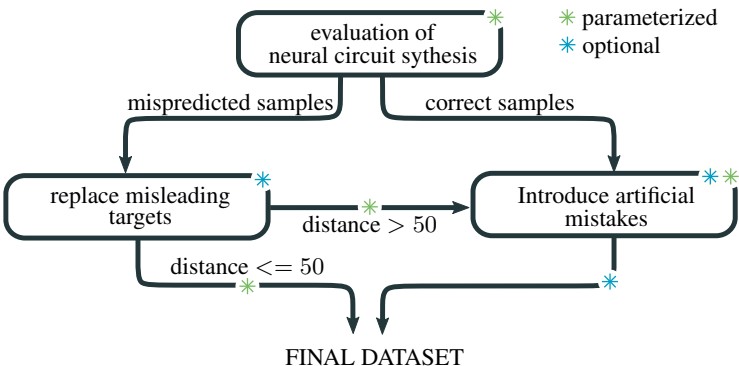

Figure 6: Overview over the data generation process.

Figure 7 shows the probability mass function for the used truncated normal distributions used in Algorithm 1. Figure 8 shows the composition of the final dataset. Samples are sorted into bins depending on the Levenshtein distance between its faulty circuit and its target circuit. While *yellow* shows all samples in the final dataset, *blue* only shows samples in the final dataset that are based on Section 3.2 and *red* only shows samples that are based on Section 3.2.

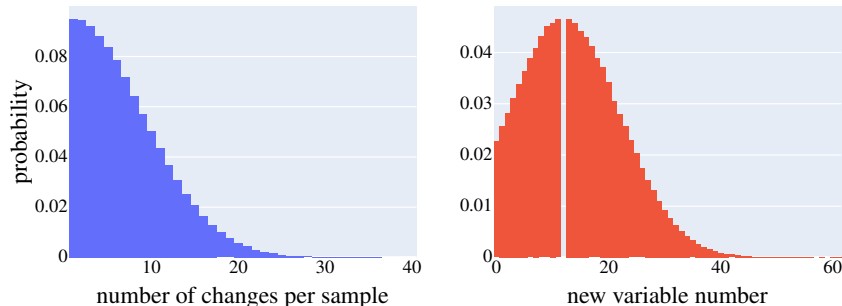

Figure 7: Probability mass function for the truncated normal distributions. Left: distribution for sampling the number of changes. Right: distribution for sampling new variable number with exemplary old variable number 12.

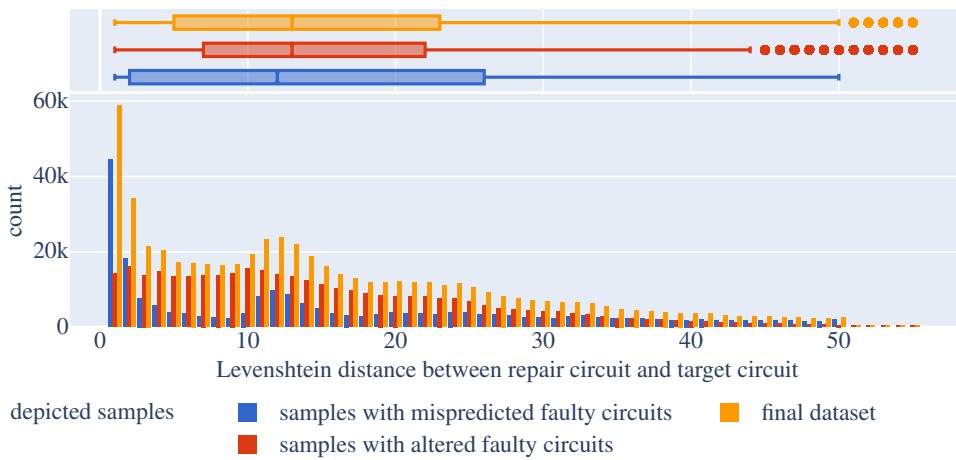

Figure 8: Composition of the final dataset. Outliers > 55 not shown.

In Figure 9, we show the composition three alternative datasets. Samples are sorted into bins depending on Levenshtein distance between its faulty circuit and its target circuit. The dataset `scpa-repair-alter-19` (*blue*) shows a dataset that is solely based on Section 3.2. Datasets `scpa-repair-gen-108` and `scpa-repair-gen-96` (*red* and *yellow*) are the two best performing datasets from all datasets we trained and based on a mixture of Section 3.2 and Section 3.2. Dataset `scpa-repair-gen-96` (yellow) is the dataset presented in this paper.

## E DIFFICULTY MEASURES

Figure 10 and Figure 11 (together with Figure 3) show predictions of the presented model, sorted into bin by specification and target size as well as Levenshtein distance between faulty input circuit and target circuit. We use beam search (beam size 16) and only display the result of the best beam. Different colors depict the different classes, a sample is sorted into, i.e. *violated* for a prediction that violates the specification and *violated (copy)* for a prediction that additionally is identical to the faulty input. *satisfied* for correct predictions and *match* for predictions that additionally are identical to the target circuit. The line shows the semantic accuracy smoothed over several bins.

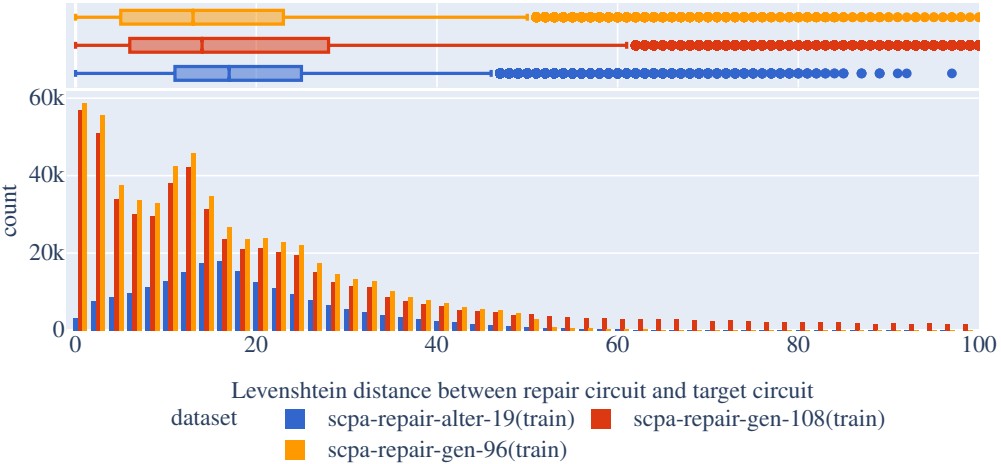

Figure 9: Comparison of the two best performing datasets and a dataset that is solely based on altered circuit data. Range 0 to 100

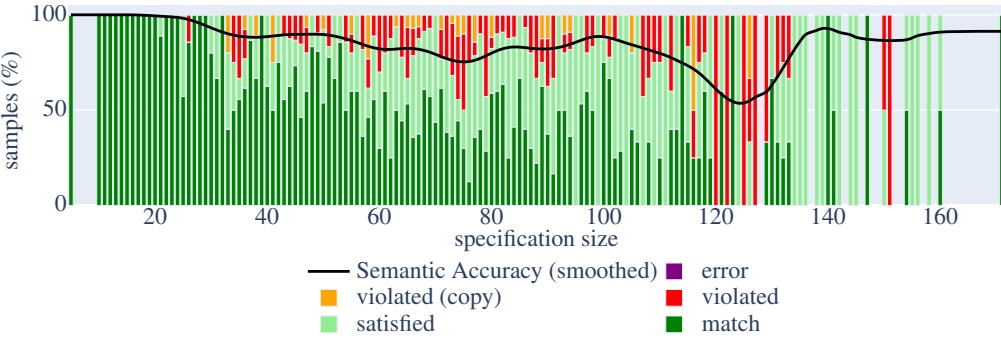

Figure 10: Accuracies and sample status broken down by the size of the specification AST.

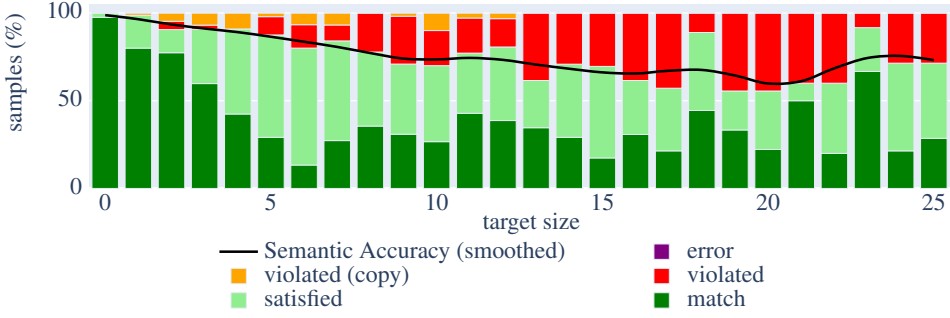

Figure 11: Accuracies and sample status broken down by the size of the target circuit (ands + latches).

## F    IMPROVEMENT MEASURES

Figure 12 shows the Levenshtein distance difference $(lev(C_p, C_t) - lev(C_f, C_t))$ between faulty input circuit and prediction. A value below zero implies syntactic improvement towards the target circuit. Figure 13 shows the number of satisfied sub-specifications. The more sub-specifications a circuit satisfies, the closer it semantically is to a correct circuit.

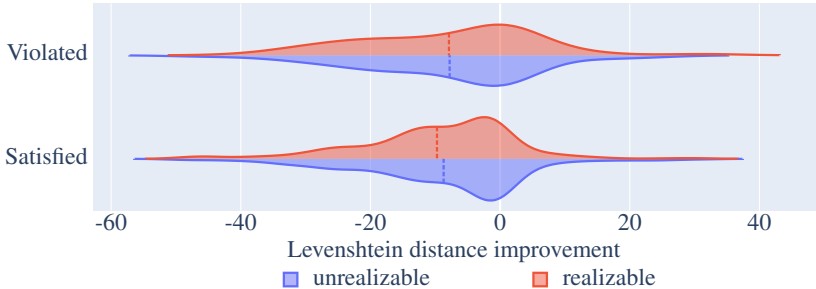

Figure 12: Violin plot of the improvement of the Levenshtein distance from the repair circuit and prediction to the target. The dashed line shows the mean of the displayed distribution.

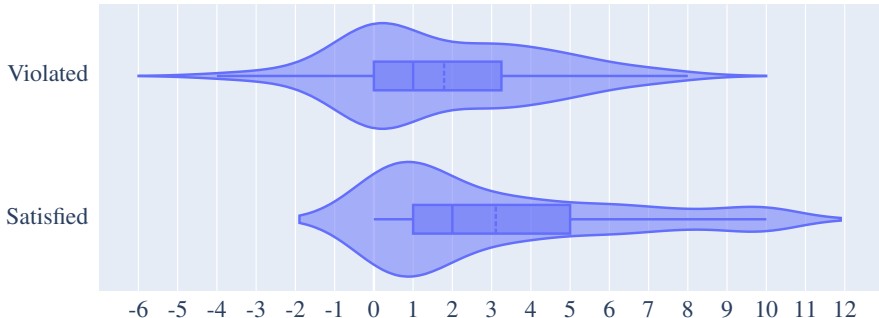

Figure 13: Violin plot of the difference between the number of sub-specs that are satisfied by the faulty input circuit vs. the predicted circuit. The larger the number the larger the improvement. Inside the violin plot is a box plot with the dashed line showing the mean of the displayed distribution. Only realizable samples.

# G ARBITER

Here, we repeat the arbiter from Figure 1, with the AIGER format for all circuits on the left of each graph representation.

```
aag 10 5 2 5 3
2           input 0 (i0)
4           input 1 (r_2)
6           input 2 (r_0)
8           input 3 (r_3)
10          input 4 (r_1)
12 18       latch 0 (l0)
14 16       latch 1 (l1)
16          output 0 (g_3)
18          output 1 (g_2)
20          output 2 (g_0)
20          output 3 (g_1)
0           output 4 (o4)
16 15 13    and-gates
18 14 13    and-gates
20 15 12    and-gates
```

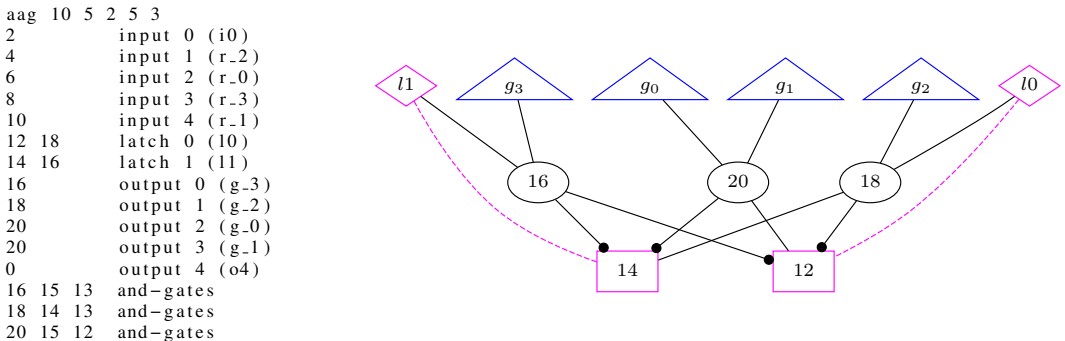

(a) Faulty circuit. Predicted in the base model (iteration 0).

```
aag 11 5 2 5 4
2           input 0 (i0)
4           input 1 (r_2)
6           input 2 (r_0)
8           input 3 (r_3)
10          input 4 (r_1)
12 13       latch 0 (l0)
14 22       latch 1 (l1)
16          output 0 (g_3)
18          output 1 (g_2)
18          output 2 (g_0)
20          output 3 (g_1)
0           output 4 (o4)
16 15 13    and-gates
18 15 12    and-gates
20 14 13    and-gates
22 19 17    and-gates
```

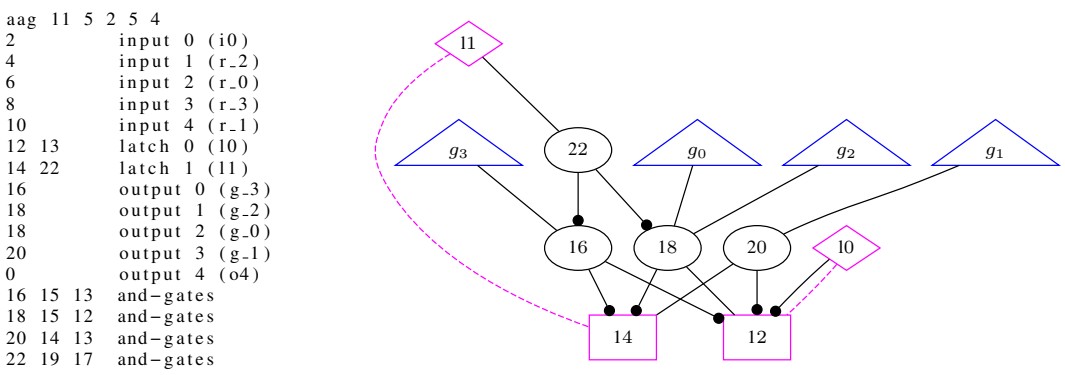

(b) Faulty circuit. Predicted in iteration 1 of the repair model.

```
            aag 12 5 2 5 5
            2
input 0 (i0)
4           input 1 (r_2)
6           input 2 (r_0)
8           input 3 (r_3)
10          input 4 (r_1)
12 13       latch 0 (l0)
14 24       latch 1 (l1)
16          output 0 (g_3)
18          output 1 (g_2)
20          output 2 (g_0)
22          output 3 (g_1)
0           output 4 (o4)
16 15 13    and-gates
18 15 12    and-gates
20 14 13    and-gates
22 14 12    and-gates
24 23 17    and-gates
```

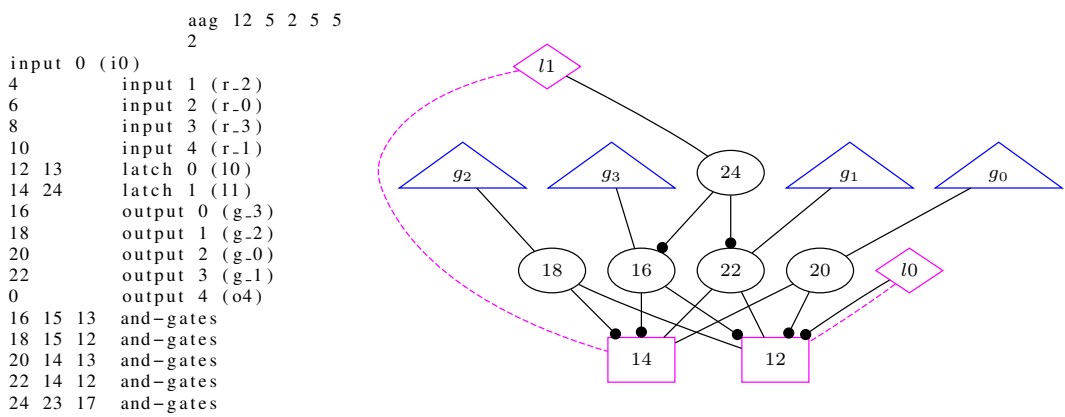

(c) Correct circuit. Predicted in iteration 2 of the repair model.

Figure 14: Failed attempt of synthesizing an arbiter and successful repair.

## H    Scaling Parameters

In this experiment, we scaled the synthesis model (Schmitt et al., 2021b) to match or exceed the number of parameters of our model. This shows that the increased number of parameters of the separated hierarchical Transformer is not the reason for the overall increase in performance. The detailed results are shown in Table 2.

Table 2: Comparison of model size and semantic accuracy between different configurations of the synthesis model and our model.

| model | parameter | sem acc. with beam size 16 |
| --- | --- | --- |
| synthesis model (baseline) | 14786372 | 77.1% |
| synthesis model: 8 local layers | 17945412 (+21.4%) | 46.3% (−30.8) |
| synthesis model: 8 global layers | 17945412 (+21.4%) | 46.5% (−30.6) |
| synthesis model: 6 encoder layers | 17945412 (+21.4%) | 58.2% (−18.9) |
| synthesis model: network size of 2048 (local layers) | 16887620 (+14.2%) | 77.4% (+0.3) |
| synthesis model: network size of 2048 (global layers) | 16887620 (+14.2%) | 77.2% (+0.1) |
| synthesis model: network size of 2048 (encoder) | 18988868 (+28.4%) | 77.3% (+0.2) |
| repair model (ours) | 17962820 (+21.5%) | 83.9% (+6.8) |

## I    Hyperparameter Study

We trained several versions of a model on the presented dataset (`scpa-repair-gen-96`) as a hyperparameter study shown in Table 3.

## J    Ablations

Figure 15 shows the semantic accuracy of an evaluation of the pipeline with a beam size of 16. If a sample has been solved correctly it will be counted as correct for all further iterations. We show the results of the model presented in this paper `exp-repair-gen-96-0` (*blue*) and two further models. The model `exp-repair-alter-19-0` (*green*) shows a model trained on a dataset that is solely based on Section 3.2. Model `exp-repair-gen-108-0` and `exp-repair-gen-96-0` (*red* and *blue*) are the two best performing models and trained on a mixture of Section 3.2 and Section 3.2. For insights into the datasets, see Figure 9.

Table 3: Hyperparameter study. Each column represents a model. The first column shows the final hyperparameters. Grey values of hyperparameters do not differ from the first column. Bold values of results show the best value in this row.

| | | | | | | | | | |
|---|---|---|---|---|---|---|---|---|---|
| parameters | | | | | | | | | |
|     embedding size | 256 | 256 | 256 | 256 | 256 | 256 | 128 | 256 | 256 |
|     network sizes | 1024 | 1024 | 1024 | 1024 | 256 | 512 | 1024 | 1024 | 256 |
|   encoder | | | | | | | | | |
|     heads global | 4 | 4 | 8 | 4 | 4 | 4 | 4 | 8 | 8 |
|     heads spec | 4 | 4 | 4 | 4 | 4 | 4 | 4 | 4 | 4 |
|     heads circuit | 4 | 4 | 4 | 4 | 4 | 4 | 4 | 4 | 4 |
|     layers global | 4 | 4 | 4 | 4 | 4 | 4 | 4 | 4 | 8 |
|     layers spec | 4 | 4 | 4 | 4 | 4 | 4 | 4 | 4 | 4 |
|     layers circuit | 4 | 6 | 4 | 4 | 4 | 4 | 4 | 8 | 8 |
|   decoder | | | | | | | | | |
|     heads | 4 | 4 | 4 | 4 | 4 | 4 | 4 | 4 | 4 |
|     layers | 8 | 6 | 8 | 10 | 8 | 8 | 8 | 8 | 8 |
| results | | | | | | | | | |
|   training split | | | | | | | | | |
|     loss | **0.03** | 0.05 | 0.03 | 0.04 | 0.04 | 0.04 | 0.04 | 0.06 | 0.08 |
|     accuracy | **0.97** | 0.95 | 0.97 | 0.97 | 0.96 | 0.96 | 0.96 | 0.94 | 0.92 |
|     (per sequence) | **0.47** | 0.40 | 0.44 | 0.43 | 0.39 | 0.42 | 0.42 | 0.32 | 0.24 |
|   validation split | | | | | | | | | |
|     loss | **0.06** | 0.13 | 0.07 | 0.06 | 0.07 | 0.06 | 0.07 | 0.14 | 0.14 |
|     accuracy | **0.95** | 0.91 | 0.95 | 0.95 | 0.95 | 0.95 | 0.95 | 0.89 | 0.88 |
|     (per sequence) | **0.32** | 0.27 | 0.30 | 0.30 | 0.29 | 0.30 | 0.30 | 0.22 | 0.19 |
|     beam size 1 | **0.59** | 0.54 | 0.58 | 0.58 | 0.53 | 0.57 | 0.53 | 0.47 | 0.42 |
|     beam size 16 | **0.82** | 0.77 | 0.81 | 0.82 | 0.80 | 0.82 | 0.81 | 0.73 | 0.69 |
|   test split | | | | | | | | | |
|     beam size 1 | 0.58 | 0.52 | 0.59 | 0.57 | 0.56 | 0.58 | **0.59** | 0.44 | 0.41 |
|     beam size 16 | 0.85 | 0.77 | **0.85** | 0.82 | 0.83 | 0.82 | 0.81 | 0.72 | 0.70 |

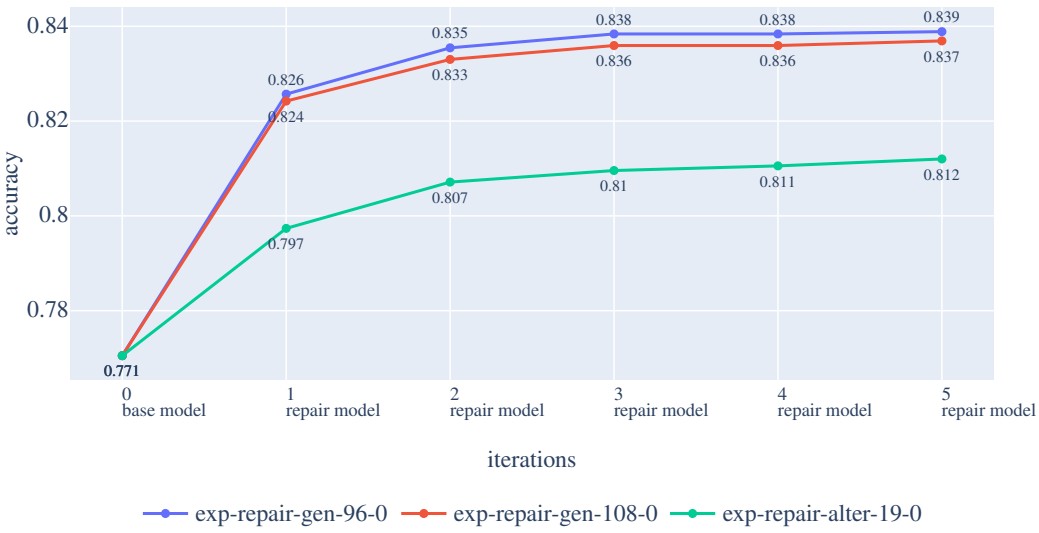

Figure 15: Accuracy after each iterative step of the pipeline (aggregated).

In Figure 16, we display all models that were trained on different datasets. For all training, we used the same hyperparameters. We plot the accuracy improvement in the pipeline, hence the performance on the reactive synthesis task on the y-axis. On the x-axis, we depict the validation accuracy, hence the performance on the circuit repair problem. Further, the pipeline accuracy improvement is based on the same distribution for all models, while the validation accuracy is based on the respective datasets used for training the model. We can see that models having a higher pipeline accuracy are trained with a dataset that included evaluation results (Section 3.2) instead of altered circuits (Section 3.2). This is not surprising, as these datasets are closer to the distribution, on which the pipeline accuracy is based. We can identify several clusters of models, of which one cluster (yellow) has relatively good validation accuracy and very good pipeline accuracy improvement. All models in this cluster improve the synthesis accuracy by more than 5 percentage points, with the highest gain of 6.8 percentage points by the model *exp-repair-gen-96-0*.

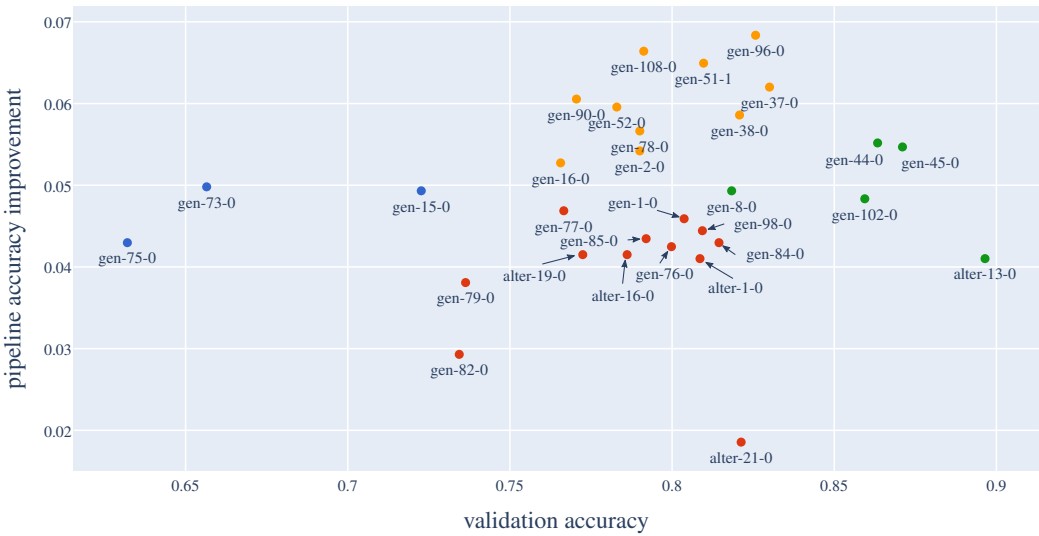

Figure 16: Pipeline accuracy improvement (percentage points) compared to validation accuracy of all trained models. Colors are based on k-means clustering with 4 clusters.

In Figure 17, we plot the mean and median of the Levenshtein distance between the faulty circuit and the target circuit for all models we trained. In the Figure 17a, the plot is dependent on the validation accuracy (the accuracy of the repair problem) and in Figure 17b the plot is dependent on the pipeline accuracy improvement (performance on the synthesis task).

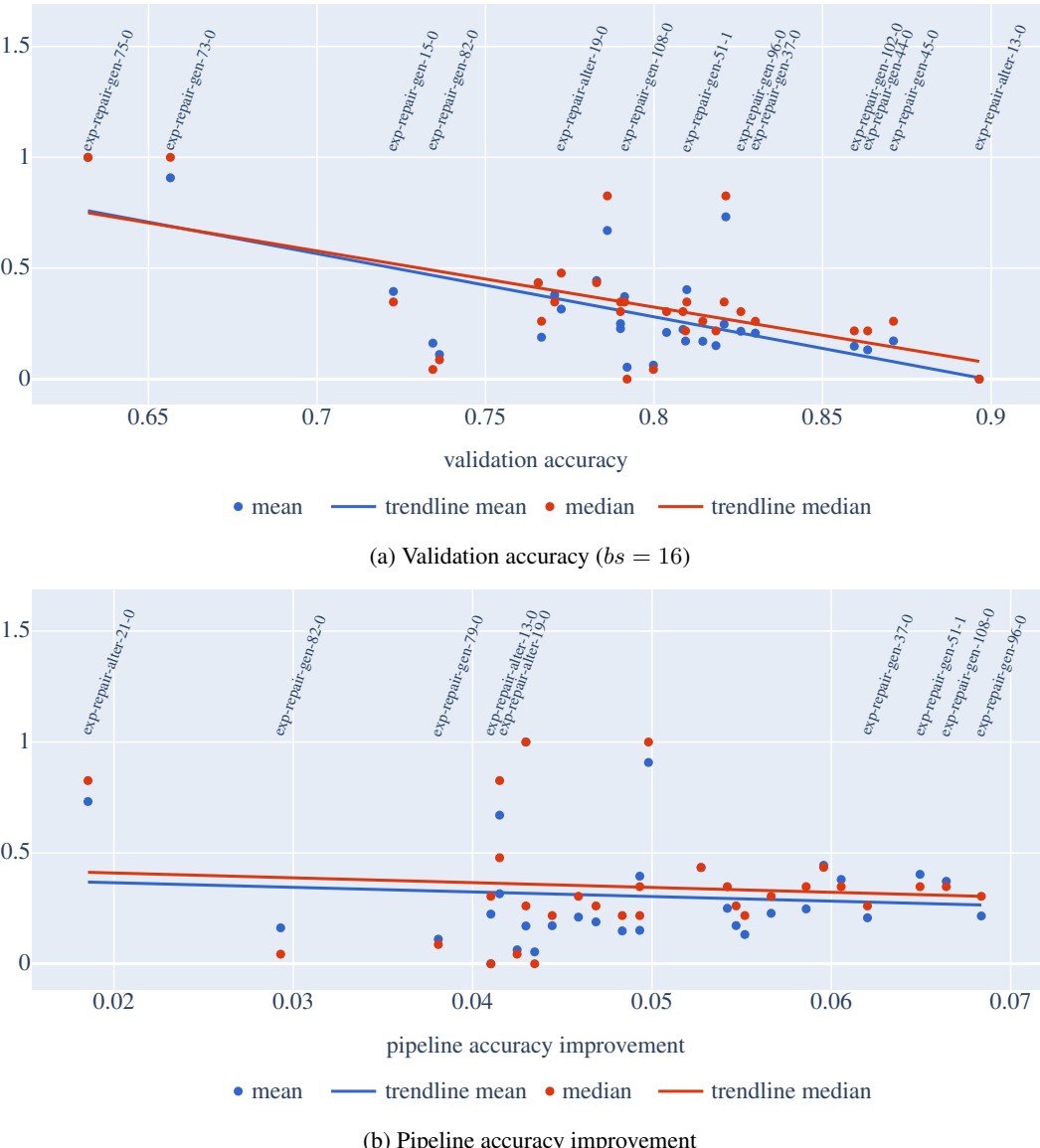

(a) Validation accuracy ($bs = 16$)

(b) Pipeline accuracy improvement

Figure 17: Validation accuracy and pipeline accuracy improvement of all trained models compared to the Levenshtein distance between faulty circuit and target circuit. Mean and median are scaled from smallest and largest values to $0$ and $1$. Only the labels of selected models are shown.

## K    SCALED DOT PRODUCT ATTENTION

For a set of queries, keys and values packed into the matrices $Q$ (queries), $K$ (keys) and $V$ (values), the scaled dot product attention (Vaswani et al., 2017) is defined as:

$$\text{Attention}(Q, K, V) = \text{softmax}(\frac{QK^T}{\sqrt{d_k}})V$$

## L  REPAIRING CIRCUITS FROM PARTIAL SPECIFICATIONS

To show the widespread applicability of our approach, we conduct another experiment. With a classical synthesis tool, we generated a test set of potentially faulty circuits from specifications where we removed the last one or two guarantees compared to the original spec. This method ensures that only 7.8% of these circuits still satisfy the original specification. We evaluated (not trained) our model on these out-of-distribution circuits and achieved 71.4% semantic accuracy after up to 5 iterations (see Table 4).

We further looked into the status of the samples broken down by (original) specification size and target size (see Figures 18 and 19). While overall results demonstrate the successful application of our model to this problem, it is noticeable that the model produces more syntax errors, most notably in larger circuits and specifications. Especially compared to Figures 10 and 11), where the model did not produce any circuits with syntax errors. This is most likely because the defective circuits in this test are out-of-distribution.

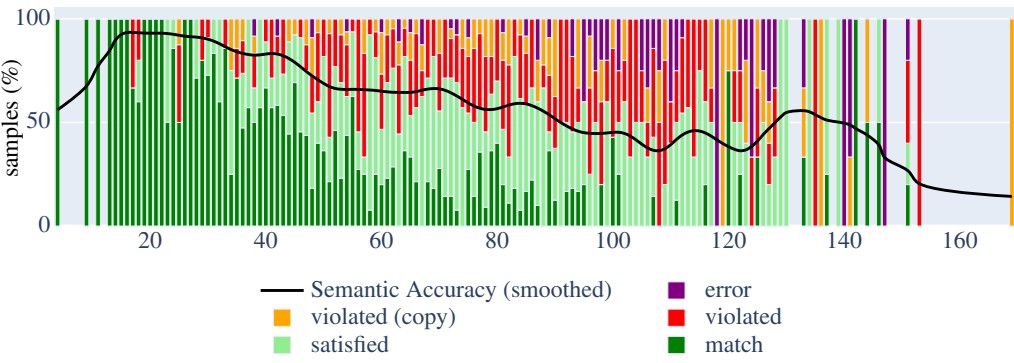

Figure 18: Accuracies and sample status broken down by the size of the specification AST. Evaluation on faulty circuits from partial specifications.

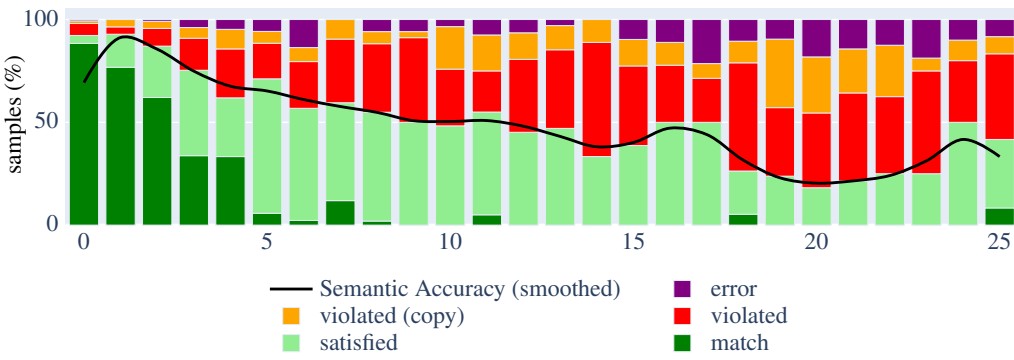

Figure 19: Accuracies and sample status broken down by the size of the target circuit. Evaluation on faulty circuits from partial specifications.

Table 4: Results on repairing faulty circuits generated with partial specifications. (Extension to Table 1)

|  | synthesis model | after first iteration | after up to $n$ iterations | $n$ |
|---|---|---|---|---|
| partial | - | 64.2% | 71.4% (+7.2) | 5 |

