# OpenReview forum: "Iterative Circuit Repair Against Formal Specifications"
_ICLR.cc/2023/Conference — ICLR 2023 poster_

### Official Review · Reviewer_NdwS · 2022-10-24

**Confidence:** 3
**Correctness:** 4
**Technical Novelty And Significance:** 2
**Empirical Novelty And Significance:** 2
**Recommendation:** 6

**Clarity, Quality, Novelty And Reproducibility:**

The paper is well written and the technique is certainly novel. I believe that the basic architecture and dataset could be reproduced from the description given in the paper.

The clarity of the baseline comparison could be improved (see weakness above).

**Strength And Weaknesses:**

# Strengths
1. The problem considered is a very important and difficult problem with a wide range of applications. Further, neural approaches make sense since in practice scalable synthesis in electronic design automation involves specializing the synthesis software by hand.

2. The technique empirically improves synthesis on established benchmarks. This is measured as Levenshtein distance in the AIGER format which intuitively captures a degree of close-ness.

# Weakness

1. The technique seems to be limited to incredibly small circuits.
    - To give context, the upper limit on 61 symbols is easily exceed by small adders (a naive adder using the popular py-aiger tool uses 65 symbols.)

2. Unconvinced by human generated error argument.
   - Having worked with AIGER a fair amount, most large circuits I create are done using tools (e.g., py-aiger) and not by hand. Thus, the argument that the introduced errors are natural seems odd.
   - A more common setting is that the specification (or unit tests) used when designing the circuit were slightly wrong, either because:
     1) It was missing assumptions.
     2) It was missing guarantees.
     3) Used bounded time horizon reasoning.
   - I would be much more interested in a neural repair that could fix this more common class of errors.

3. Perhaps I missed something, but there seems to be a missing comparison with non-neural techniques. As the authors point out, this is a very mature field, so it would be have been nice to see a detailed comparison.

**Summary Of The Paper:**

This paper proposes using a neural network to iteratively repair sequential circuits represented as And Inverter Graphs (AIGs) against temporal logic specifications. The architecture is a variation of a transformed that separately takes as input the circuit and the specification. The specification is also separated into its assumptions and its guarantees. The network is then trained by (i) sampling circuits and specifications from well established benchmarks (ii) randomly introducing errors by changing the textual representation of the AIG.

Empirical results suggest that this technique is indeed able to correct incorrectly synthesized specifications and that repeated applications further improve performance. Importantly this can be used in combination with neural network based synthesis of the initial circuit.

**Summary Of The Review:**

Overall, I believe the future potential of the propose method out weight the clear limitations on scale and questionable motivation for the error insertion. Circuit repair (and circuit synthesis) are computationally difficult problems in general and the field largely relies on hand tuned heuristics. Supporting data-driven techniques is a natural means to accelerate synthesis in specific target domains.

---

> ### Author Response · Authors · 2022-11-12
> **Reply to Reviewer NdwS**
>
> We thank the reviewer for taking the time to review our submission and for their valuable suggestions.
>
> - "The technique seems to be limited to incredibly small circuits."
> As the reviewer mentioned in his review, the formal circuit repair problem (with the synthesis problem being a special case) is a computationally hard problem (2-EXPTIME-complete for LTL formulas).
> This naturally limits the specifications and the circuits that can be currently handled. With the combination of neural and classical approaches, we expect a significant leap in the near future.
> Still, the circuits considered in this paper are comparable to Syntcomp specs: 242 of 346 have less than 6 inputs and 274 have less than 6 outputs. We plan to lift this limitation in the future with additional computational resources.
>
>
> - "A more common setting is that the specification (or unit tests) used when designing the circuit were slightly wrong, either because 1. It was missing assumptions. 2. It was missing guarantees. 3. Used bounded time horizon reasoning."
> We thank the reviewer for these very interesting suggestions! We conducted a preliminary experiment to validate that our approach is valuable when encountering, for example, missing guarantees (the reviewer's second point). With a classical synthesis tool, we generated a test set of potentially faulty circuits from specifications where we removed the last one or two guarantees compared to the original spec (only 7.8% of these circuits still satisfy the original spec). We evaluated (not trained!) our model on these circuits and achieved 64.2% semantic accuracy.
>
> - "Perhaps I missed something, but there seems to be a missing comparison with non-neural techniques. As the authors point out, this is a very mature field, so it would be have been nice to see a detailed comparison."
> We would like to emphasize that we think that a neural approach to such formal problems is orthogonal to classical approaches (see timeout dataset). They truly shine by combining the best of both worlds: efficiency by using the neural network as a heuristic and completeness by applying model checking and using classical synthesis as a fall-back.
> That being said, we acknowledge the reviewer's interest in this topic. We expanded the related work section on this topic and are currently running a few comparison experiments to provide more insights. We will provide the results as soon as possible.
>
> Please let us know if you have additional questions or concerns.

---

> > ### Author Response · Authors · 2022-11-18
> > **Follow-up experiments on timeouts and faulty circuits generated with missing guarantees**
> >
> > Dear reviewer NdwS
> >
> > We ran follow-up experiments on faulty circuits generated with missing guarantees. We applied our model iteratively, where we could further improve to 71.4% (+ 7.2 percentage points). We updated these results in the paper and added a new Appendix section L.
> >
> > Additionally, we ran experiments comparing our approach to the classical methods.
> > We compare against the classical synthesis tool Strix, which is the leading tool for years for LTL-synthesis. The dataset ```timeouts``` contains specifications Strix could not synthesize in 120s. For these samples, our model could produce 37.6% correct solutions. We created an additional dataset, ```timeouts-1h```, containing specifications that Strix could not solve with a timeout of 1 hour. On these samples, our model still achieved 34.2% correct solutions.
> >
> > For convenience, we pasted our key results, including the new experiments below. The table is also updated in the paper.
> >
> >
> > |                                   | synthesis model | after first iteration | after up to n iterations | n |
> > |-----------------------------------|-----------------|-----------------------|----------------------------|-----|
> > | ```test``` (repair dataset)    | -               |  84.2%             | 87.5% (+3.3)          | 5   |
> > | ```test``` (synthesis dataset) |  77.1%      |  82.6% (+5.5)   | 83.9% (+6.8)          | 5   |
> > | ```timeouts``` (120s)          |  30.7%       |  34.9% (+4.2)    | 37.6% (+6.9)       | 5   |
> > | ```timeouts-1h``` (1h)            | 26.1%        | 32.5% (+7.4)      |34.2% (+8.1)        |5
> > | ```syntcomp```                 |  64.1%       |  71.7% (+7.6)    | 75.9% (+11.8)       | 2   |
> > | ```smart home```               |  42.9%       |  61.9% (+19)     | 66.7% (+23.8)       | 2   |
> > | ```partial``` (repair)                  | -               | 64.2%                 | 71.4%  (+7.2)                     | 5   |

---

### Official Review · Reviewer_LxSj · 2022-10-25

**Confidence:** 3
**Correctness:** 3
**Technical Novelty And Significance:** 3
**Empirical Novelty And Significance:** 3
**Recommendation:** 6

**Clarity, Quality, Novelty And Reproducibility:**

The paper is clearly written and easy to follow. The approach looks novel to me. The authors promised the release of datasets and code for reproducibility.

**Strength And Weaknesses:**

Strength:

+ The approach is technically sound. While there have been a variety of work on generating logical formulas, formal specifications, or formal proofs with language models these days, the problem this paper is targeting (with language models) looks novel to me.

+ The empirical evaluation is extensive, and validates the proposed framework. The difficulty measures and Levenshtein distance help better understand the capability of language models on the task.

Weakness:

- The specifications are restricted to 5 inputs and 5 outputs, no more than 12 properties. Did you observe a degradation in performance as the size of specifications increases?

- What are the advantages of hierarchical transformers over vanilla ones? Although the experiments are extensive, I can't seem to find this particular dimension of evaluation. Also, would fine-tuning a pre-trained language model be as good as training a hierarchical transformer from scratch?

**Summary Of The Paper:**

The paper investigates the application of transformer models to repairing circuits given formal specifications. The transformer takes a faulty circuit represented in AIGER format and a specification in LTL, and outputs a circuit that hopefully satisfies the specification. The paper also comes with a detailed description of the generation process of datasets.

**Summary Of The Review:**

The framework is well-designed and the performance looks good. I lean toward accepting the paper.

---

> ### Author Response · Authors · 2022-11-12
> **Reply to Reviewer LxSj**
>
> We thank the reviewer for taking the time to review our submission and for their valuable feedback.
>
> - "The specifications are restricted to 5 inputs and 5 outputs, no more than 12 properties. Did you observe a degradation in performance as the size of specifications increases?"
> We refer the reviewer to Figure 11 in the appendix; a degradation in performance with increasing specification size is recognizable, although it is not as significant as expected from the synthesis problem's complexity (2-EXPTIME). This speaks for the strong generalization capabilities of the Transformer on this problem.
> We capped our datasets at 5 inputs/outputs and 12 properties to stay comparable to prior work and to fit our resource restrictions. These restrictions can be lifted with additional computing resources.
>
> - "What are the advantages of hierarchical transformers over vanilla ones? Although the experiments are extensive, I can't seem to find this particular dimension of evaluation."
> Overall, we consistently observed a higher accuracy for the hierarchical Transformer compared to a vanilla Transformer with the same parameter size: 31.5% vs. 32.4% syntactic accuracy on the validation set, 83.9% vs. 84.8% semantic accuracy on the test set, and 71.7% vs. 73.8% semantic accuracy on the Syntcomp dataset; which is why we stuck to the hierarchical Transformer. Additionally, the hierarchical Transformer is (in contrast to the vanilla transformer) invariant against the order of the assumptions and guarantees in the specification.
>
> - "Also, would fine-tuning a pre-trained language model be as good as training a hierarchical transformer from scratch?"
> We haven't conducted this experiment. We are also interested in comparing a pre-trained vanilla transformer and a pre-trained hierarchical transformer. We postpone this to future work and thank the reviewer for their suggestion.
>
> Please let us know if you have additional questions or concerns.

---

### Official Review · Reviewer_ocom · 2022-10-25

**Confidence:** 2
**Correctness:** 4
**Technical Novelty And Significance:** 2
**Empirical Novelty And Significance:** 2
**Recommendation:** 5

**Clarity, Quality, Novelty And Reproducibility:**

The paper  is not entirely self contained with many of the details given in the
Appendix. Though it would be impossible to fit everything in the page limit, I
think that some additional details on major concepts could be given; for
instance, the reactive synthesis problem is never defined and the term
'attention' is used a number of times but never explained. The novelty of the
paper is not particularly strong as it is highly incremental to Le at al.
(2021).

**Strength And Weaknesses:**

+ Good empirical analysis that shows the efficacy of the proposed method in
circuit repair and its superior performance in reactive synthesis when compared
with the state-of-the-art.

- Highly incremental to previous work (the model proposed is a variation of the
  work from Li et al. (2021) and the data generation algorithm is a standard
  fault injection algorithm).

- Some comparisons with related work are shallow. For instance, the
  advantages/disadvantages of the present method to symbolic synthesis are not
  discussed.

**Summary Of The Paper:**

The paper extends hierarchical transformers (neural network models for
synthesising circuits from linear time specifications)  to build models  for
the repair of circuits so that they satisfy a certain linear time property. The
models, which are also used in an iterative procedure to do reactive synthesis,
are trained on a novel algorithm for introducing faults to a dataset.

**Summary Of The Review:**

The paper introduces neural models for circuit synthesis which exhibit gains
over the state-of-the-art, albeit being highly incremental to previous work.

---

> ### Author Response · Authors · 2022-11-12
> **Reply to Reviewer ocom**
>
> We thank the reviewer for taking the time to review our submission and for their helpful suggestions.
>
> - “The novelty of the paper is not particularly strong as it is highly incremental to Li at al. (2021).”
> We want to emphasize that the motivation and contribution of this paper is driven by application: the formal circuit repair problem. Therefore, the novelty of the paper is not limited to the architecture; our adjusted hierarchical Transformer is also a means to an end here.
> We strongly believe that our findings will help in applying transformer models in other symbolic domains with multi-modal input.
>
> - “the data generation algorithm is a standard fault injection algorithm.”
> We are not aware of a “standard” fault injection algorithm for low-level circuit designs (there are works on vhdl-level, e.g., [1]). We don’t see being straightforward here as a particularly bad thing, as it makes the approach applicable to any hardware design.
>
> - “Some comparisons with related work are shallow. For instance, the advantages/disadvantages of the present method to symbolic synthesis are not discussed.”
> The deep learning synthesis+repair vs. the classical synthesis+repair approach is essentially a trade-off between efficiency and completeness. While the Transformer provides many possibly correct predictions (that can be checked easily) very fast, it is not complete. Vice versa, the classical tools are complete but very slow. They are orthogonal approaches (see timeout dataset). We expanded the related work section on this topic.
>
> - “I think that some additional details on major concepts could be given; for instance, the reactive synthesis problem is never defined and the term ‘attention’ is used a number of times but never explained.”
> We added a definition of attention, as defined by Vaswani et al. to Appendix J and the definition of the reactive synthesis problem to Appendix A.
>
> Please let us know if you have additional questions or concerns.
>
> ---
> [1] Lala, Parag K. “Transient and permanent fault injection in VHDL description of digital circuits.” (2012).

---

### Official Review · Reviewer_Y3T4 · 2022-10-31

**Confidence:** 3
**Correctness:** 3
**Technical Novelty And Significance:** 2
**Empirical Novelty And Significance:** 2
**Recommendation:** 5

**Clarity, Quality, Novelty And Reproducibility:**

The paper is mostly well written and straightforward to follow.  The exception are the figures, which would benefit from more information in the captions.

The novelty of this paper is limited.  The main contributions are a modest variant on an existing hierarchical transformer architecture, and a straightforward data augmentation procedure.  Individually or together these do not seem substantially novel, but that by itself is not necessarily a problem.

The description is precise enough to attempt to reproduce this work.  The authors state in several places that they will produce the source code for this work, which will improve reproducitibility further.

### Questions

- Clarify “filtered out samples that exceed our model restrictions”
- It’s not clear whether “after last iteration” in Table 1 means after exactly 2 iterations of after some arbitrary number.
- In Figure 4 has an “error” label purple, but I do not see any corresponding purple in the graph itself.  Is this just too small to see on the graph?
- Clarify “accuracy” in Figures as semantic accuracy and provide explicit meaning to [0, 1] range.

**Strength And Weaknesses:**

### Main strengths:

- Well motivated problem
- Mostly well written and easy to follow
- Comprehensive experimental analysis (with some caveats)

### Main weaknesses:

- Most of the value of this contribution rests upon whether the following causal claims are true and well-justified: the new architecture and/or the data-augmentation procedure caused the improvements in performance of state of the art.  Despite a number of different experimental analyses in the paper, determining this is not straightforward.  In particular, I cannot see where if anywhere the number of parameters are controlled for.  The paper does say that using separated heads leads to an increase in the number of parameters, but I do not see any evidence in this paper to suggest that performance increases over previous methods is not attributed simply to this network being larger.  Also, the experimental results do not allow us to distinguish whether improvements are from the model changes or from the data changes.
- I have some reservations about the use of Levenshtein distance as a metric for the quality of a synthesized circuit.  Obviously there is no “right” metric, and Levenshtein may be used in prior work, but a more semantic property could be used in addition to convey improvement.  For instance, one could try to look at (some approximation of) the increase or decrease in the number of satisfying traces

**Summary Of The Paper:**

Given a formal specification in LTL, this paper introduces a transformer architecture that aims to transform a defective circuit into a repaired one, in accordance to the spec.   The primary contribution is in the transformer neural architecture, which they call the separated hierarchical transformer since it has separate local layers that do not share model parameters.

In addition, they introduce a data generation procedure that models human errors to encourage generalization to more complex specifications.

**Summary Of The Review:**

- Well written paper with comprehensive experimental analysis
- Limited novelty
- Would like to see much more evidence that claimed contributions are responsible for improved performance

---

> ### Author Response · Authors · 2022-11-12
> **Reply to Reviewer Y3T4**
>
> We thank the reviewer for taking the time to review our submission and for their valuable feedback.
>
>  - Regarding “cannot see where if anywhere the number of parameters are controlled for [...and..] I do not see any evidence in this paper to suggest that performance increases over previous methods is not attributed simply to this network being larger.”
> We thank the reviewer for this suggestion. We ran experiments increasing the model size of the previous method (see table below) up to 28.4%. This shows that the performance increase of our method is not simply attributed to the network being larger. We added the results in Appendix I.
>
> | model                |parameter          | beam search 16 sem acc  |
> |----------------------|--------------------|---------------|
> | synthesis model (base)           | 14786372           | 77.1%         |
> | synthesis model: 8 local layers   | 17945412 (+ 21.4%) | 46.3% (-30.8) |
> | synthesis model: 8 global layers  | 17945412 (+ 21.4%) | 46.5% (-30.6) |
> | synthesis model: 6 encoder layers | 17945412 (+ 21.4%) | 58.2% (-18.9) |
> | synthesis model: 2048 feed-forward-size local layers | 16887620 (+ 14.2%) | 77.4% (+0.3)  |
> | synthesis model: 2048 feed-forward-size global layers   | 16887620 (+ 14.2%) | 77.2% (+0.1)  |
> | synthesis model: 2048 feed-forward-size encder  | 18988868 (+ 28.4%) | 77.3% (+0.2)  |
> | repair model (ours)        | 17962820 (+ 21.5%) | 83.9% (+6.8)  |
>
> - Regarding “I have some reservations about the use of Levenshtein distance as a metric for the quality of a synthesized circuit. Obviously there is no “right” metric, [...] but a more semantic property could be used in addition to convey improvement. For instance, one could try to look at (some approximation of) the increase or decrease in the number of satisfying traces”
> We thank the reviewer for this interesting suggestion.
> We would 1) like to explain why such a trace-based semantic metric (even approximations) is not directly applicable in our setting, which was the reason we went for the (imperfect) Levenshtein distance, before 2) providing a semantic distance check inspired by the reviewer’s suggestion.
>     1. a) faults introduced by the algorithm (and also mistakes by humans) can result in syntactically incorrect circuits, which cannot be simulated
>     b) a sequential circuit is usually satisfied by infinitely many traces
>     c) a prediction has not necessarily the same realizability label as the target circuit. If, for example, the network mispredicts a circuit to be unrealizable the simulated circuit will output traces for the counterstrategy instead.
>     2. That being said, we came up with the following semantic metric and did an initial experiment for realizable specs only:
> We deleted a guarantee from the specification and model-checked the faulty circuit and the predicted circuit. We repeated this for every guarantee in the specification. This metric is well-suited and confirms our Levenshtein experiments: In 66.3%, the prediction satisfied more sub-specs, and in 0.9%, the prediction satisfied fewer sub-specs. In 32.7%, the prediction did not improve according to this metric. We will describe this alternative metric and the experiment in the paper’s appendix. The Levenshtein distance still has the benefit of being easy to implement and being applicable to many domains.
>
> - “[...] figures, which would benefit from more information in the captions.”
> We are restricted by the page limit. If the reviewer has specific suggestions, we will gladly try implementing them.
>
> - “Clarify “filtered out samples that exceed our model restrictions””
> To be comparable to previous work (and to meet our hardware constraints), we filtered specifications with more than 5 inputs/outputs, more than 12 properties, and properties of size greater than 25. After filtering, 145 out of 346 formulas from the Syntcomp dataset remain. This limitation can be lifted in the future with additional computational resources.
>
> - “It’s not clear whether “after last iteration” in Table 1 means after exactly 2 iterations of after some arbitrary number.”
> For the timeouts and the testset, we kept only the best beam after each iteration and did 5 iterations. For the Syntcomp and smart home dataset, we did 2 iterations on every beam, which performed better than performing 5 iterations on a single beam. Thanks for pointing this out; we will add this information to the paper.
>
> - “In Figure 4 has an “error” label purple, but I do not see any corresponding purple in the graph itself. Is this just too small to see on the graph?”
> We added this label to track errors of external tools, such as the model checker. However, no instance was labeled with an error. We will delete the label from the plot for the final version.
>
> - “Clarify “accuracy” in Figures as semantic accuracy and provide explicit meaning to [0, 1] range.”
> Thanks, we will make this clear in the paper.
>
> Please let us know if you have additional questions or concerns.

---

> > ### Author Response · Authors · 2022-11-15
> > **New Revision**
> >
> > As promised, we have now updated the paper with the reviewer's suggestions. We also extended the experiment regarding a more semantic metric, which can now be found in Appendix K.

---

### Author Response · Authors · 2022-11-18
**Summary of Changes**

* Added a note to the related work section on differences between classical and neural approaches.
* Added a symbolic metric to measure improvement (Appendix K, Section 5.2)
* Extended captions for Figures and better axis titles (Figures 4, 5, 11, 12)
* Fixed error label (Figure 4)
* Scaled evaluations on timeouts dataset to samples that Strix could not solve within 1 hour. (Table 1, Sections 5.3)
* Elaborated on our input restrictions (Sections 5.1 and 5.3)
* Added definition of Reactive Synthesis (Appendix A)
* Parameter comparison to show that the increased number of parameters of the separated hierarchical Transformer is not the reason for the overall increase in performance (Appendix I)
* Added experiments on repairing faulty circuits from partial specifications (Appendix L)
* Definition of scaled dot product attention added (Appendix J)

---

### Decision · Program_Chairs · 2023-01-20

**Decision:**

Accept: poster

**Justification For Why Not Higher Score:**

This problem is specialized enough that it might not yet be of interest to a larger subset of the community.

**Justification For Why Not Lower Score:**

The main result makes an interesting advance towards a difficult problem.

**Metareview: Summary, Strengths And Weaknesses:**

This paper introduces a way to repair sequential circuits that do not meet a given LTL specification. The overall idea follows closely previous work on synthesis from LTL specifications [1], except that the failing circuit becomes input to the Transformer. The paper also introduces a variant on the hierarchical Transformer architecture and a data augmentation procedure.

The main result is that using deep learning can be effectively used to predict the design of a correct circuit given a failing circuit and its LTL specification, and this results in correctly synthesizing many more circuits. This is indeed an interesting result, and could turn out to be an important next step in a difficult problem.

The architectural and learning innovations here are fairly straightforward, as acknowledged by several of the reviewers, but the main innovation here is in evaluating synthesize+repair versus a synthesize-only neural workflow.

The most serious weaknesses discussed by the reviewers were adequately addressed in the author response. These were:

   * Controlling for the number of parameters (Y3T4): Additional experiments in the response addressed this.
   * Comparison to symbolic methods (oocom, NdwS): This is well addressed by the Strix comparison, which is impressive.
   * Advantages compared to hierarchical transformers compared to vanilla: This is also addressed in the response to LxSj.

I would emphasize to the authors, however, that many of these results (even ones that were in the original draft) are relegated to appendices. These are important comparisons, as can be seen from the fact that several reviewers asked for them. I would urge the authors to highlight all three of the results above in the main paper. They are more important in my view than (say) Figure 3, which could be safely relegated to an appendix.

A final weakness of the paper is that the specifications here are extremely small, and would not cover many real world circuits. The authors responding by pointing out that the synthesis problem is 2-EXPTIME. That would usually be a good response, but it is not a good response here. The problem is that while the exact problem is indeed intractable, but the neural method does not solve the exact problem, because its output is not guaranteed to meet the specification (of course you can use it to generate-and-test with a verifier afterwards). Part of the point of using a deep learning method for synthesis is to give up the formal guarantees to get scalability.

However, I am persuaded by the fact that: (a) the standard benchmarks are necessarily going to be restricted to those that symbolic methods can solve, and (b) the overall idea of synthesize+repair using Transformers can in principle be applied to much larger circuits (if one resorts to empirically verifying the circuits rather than formal verification). But it would be good if a more nuanced discussion of this point appeared in the paper.

[1] Frederik Schmitt, Christopher Hahn, Markus N. Rabe, and Bernd Finkbeiner. Neural circuit synthesis from specification patterns. NeurIPS 2021.




**Note From Pc:**

if the above contains the word "oral" or "spotlight" please see: "oral" presentation means -> notable-top-5% and "spotlight" means -> notable-top-25%. As stated in our emails, we are disassociating presentation type from AC recommendations